# A Systematic Review on the Molecular Mechanisms of Resveratrol in Protecting Against Osteoporosis

**DOI:** 10.3390/ijms26072893

**Published:** 2025-03-22

**Authors:** Ahmad Nazrun Shuid, Nurul Alimah Abdul Nasir, Norasikin Ab Azis, Ahmad Naqib Shuid, Norhafiza Razali, Haryati Ahmad Hairi, Mohd Fairudz Mohd Miswan, Isa Naina Mohamed

**Affiliations:** 1Department of Pharmacology, Faculty of Medicine, Universiti Teknologi MARA, Sungai Buloh Campus, Jalan Hospital, Sungai Buloh 47000, Selangor, Malaysia; anazrun@uitm.edu.my (A.N.S.); nurulalimah@uitm.edu.my (N.A.A.N.); norasikin_abazis@uitm.edu.my (N.A.A.); norhafiza8409@uitm.edu.my (N.R.); 2Advanced Medical and Dental Institute, Universiti Sains Malaysia, Bertam, Kepala Batas 13200, Penang, Malaysia; naqib@usm.my; 3Department of Biochemistry, Faculty of Medicine, Manipal University College Malaysia, Jalan Batu Hampar, Bukit Baru 75150, Melaka, Malaysia; haryatiahmadhairi@gmail.com; 4Department of Orthopaedics, Faculty of Medicine, Universiti Teknologi MARA, Sungai Buloh Campus, Jalan Hospital, Sungai Buloh 47000, Selangor Darul Ehsan, Malaysia; fairudz@uitm.edu.my; 5Department of Pharmacology, Faculty of Medicine, Universiti Kebangsaan Malaysia, Jalan Yaacob Latif, Cheras 56000, Kuala Lumpur, Malaysia

**Keywords:** resveratrol, osteoporosis, osteoblast, molecular mechanisms, bone health

## Abstract

Osteoporosis is a prevalent metabolic bone disorder characterized by decreased bone mineral density and increased fracture risk, particularly among aging populations. While conventional pharmacological treatments exist, they often have adverse effects, necessitating the search for alternative therapies. Resveratrol, a naturally occurring polyphenol, has gained significant attention for its potential osteoprotective properties through various molecular mechanisms. This systematic review aims to comprehensively analyze the molecular pathways through which resveratrol protects against osteoporosis. Using an advanced search strategy in the Scopus, PubMed, and Web of Science databases, we identified 513 potentially relevant articles. After title and abstract screening, followed by full-text review, 28 studies met the inclusion criteria. The selected studies comprised 14 in vitro studies, 8 mixed in vitro and in vivo studies, 6 in vivo studies, and 1 cross-sectional study in postmenopausal women. Our findings indicate that resveratrol exerts its osteoprotective effects by enhancing osteoblast differentiation through the activation of the Phosphoinositide 3-Kinase/Protein Kinase B (PI3K/Akt), Sirtuin 1 (SIRT1), AMP-Activated Protein Kinase (AMPK), and GATA Binding Protein 1 (GATA-1) pathways while simultaneously inhibiting osteoclastogenesis by suppressing Mitogen-Activated Protein Kinase (MAPK) and TNF Receptor-Associated Factor 6/Transforming Growth Factor-β-Activated Kinase 1 (TRAF6/TAK1). Additionally, resveratrol mitigates oxidative stress and inflammation-induced bone loss by activating the Hippo Signaling Pathway/Yes-Associated Protein (Hippo/YAP) and Nuclear Factor Erythroid 2-Related Factor 2 (NRF2) pathways and suppressing Reactive Oxygen Species/Hypoxia-Inducible Factor-1 Alpha (ROS/HIF-1α) and NADPH Oxidase 4/Nuclear Factor Kappa-Light-Chain-Enhancer of Activated B Cells (Nox4/NF-κB). Despite promising preclinical findings, the low bioavailability of resveratrol remains a significant challenge, highlighting the need for novel delivery strategies to improve its therapeutic potential. This review provides critical insights into the molecular mechanisms of resveratrol in bone health, supporting its potential as a natural alternative for osteoporosis prevention and treatment. Further clinical studies are required to validate its efficacy and establish optimal dosing strategies.

## 1. Introduction

Osteoporosis is a chronic metabolic bone disorder characterized by decreased bone mineral density (BMD) and structural deterioration, leading to an increased risk of fractures. Affecting millions of individuals worldwide, osteoporosis is particularly prevalent among postmenopausal women and the elderly due to hormonal changes, aging-related bone loss, and lifestyle factors [1]. Conventional pharmacological treatments, such as bisphosphonates, selective estrogen receptor modulators (SERMs), and monoclonal antibodies, have demonstrated efficacy in slowing bone loss but are often associated with adverse effects, including gastrointestinal discomfort and increased risks of atypical fractures [2]. Consequently, there is a growing interest in natural compounds, such as polyphenols, that possess osteoprotective properties with fewer side effects. Among these, resveratrol, a naturally occurring polyphenol found in grapes, berries, and peanuts, has emerged as a promising bioactive compound with potential benefits for bone health [3].

Resveratrol exerts its osteoprotective effects through multiple molecular mechanisms, including the activation of the SIRT1 signaling pathway, the modulation of oxidative stress, and the regulation of osteoblast and osteoclast activity [4]. Studies have demonstrated that resveratrol enhances osteoblast differentiation by upregulating osteogenic markers such as runt-related transcription factor 2 (RUNX2), alkaline phosphatase (ALP), and osteocalcin [5]. Simultaneously, it inhibits osteoclastogenesis by suppressing receptor activator of nuclear factor-kappa B ligand (RANKL) signaling, thereby reducing bone resorption [6]. Additionally, resveratrol has been shown to mitigate oxidative stress-induced bone loss by enhancing antioxidant enzyme activity and reducing reactive oxygen species (ROS) levels, which are known to accelerate bone degradation [7]. These molecular actions highlight resveratrol’s potential as a therapeutic agent for osteoporosis prevention and treatment.

Despite promising preclinical and in vitro findings, challenges remain in translating resveratrol’s osteoprotective effects into clinical applications due to its chemical instability, poor water solubility, low bioavailability, and rapid metabolism in the intestine and liver [8,9]. Several strategies, including nanoparticle-based delivery systems and structural modifications, are being explored to enhance its bio efficacy [10]. Moreover, while some clinical studies have reported improved bone markers and BMD following resveratrol supplementation, further large-scale, randomized controlled trials (RCTs) are necessary to establish its long-term efficacy and safety [11]. This systematic review aims to comprehensively analyze the molecular mechanisms by which resveratrol protects against osteoporosis, providing insights into its potential as a natural therapeutic agent for bone health.

## 2. Methods

### 2.1. Search Strategy

The search strategy was developed using the PICO framework (Population, Intervention, Comparison, and Outcome) to structure the research question effectively. The goal was to identify relevant studies investigating the molecular mechanisms of resveratrol in protecting against osteoporosis.

The search strategy was conducted by ANS and NAAN in accordance with the Meta-Analysis and Systematic Reviews of Observational Studies (MOOSE) guideline [12] and the Preferred Reporting Items for Systematic Review and Meta-Analysis Protocols (PRISMA-P) [13].

A systematic review of the literature was conducted across the Scopus, PubMed, and Web of Science databases using Medical Subjects Headings (MeSH) indexes and keyword searches. The string used included the entry term “Resveratrol”, which was searched in combination (AND) with the terms “osteo*” OR “bone”. The same combination of terms was used for all the databases. The search was limited to articles published within the past five years (2020–2025) to focus on the most recent literature, considering the rapid evolution of scientific knowledge in the field of nutraceuticals.

The inclusion criteria included the following:Peer-reviewed journal articles;Studies on resveratrol’s effects on bone that assess molecular parameters to determine its mechanisms in protecting against osteoporosis.

The exclusion criteria included the following:Articles not published in English due to limited resources for translation;Reviews or meta-analyses without primary data;Letters, editorials, or case studies;Studies where resveratrol was combined with other agents or special carriers, making it difficult to determine its independent contribution to the mechanism;Studies that did not measure molecular parameters.

### 2.2. Selection Criteria

All eligible studies were selected following the PICOS (Population, Intervention, Comparison/Comparator, Outcomes, and Study design) model (Table 1) [14].

First, the articles retrieved from the databases were refined using the provided tools. The titles of the selected articles from the three databases were compared, and any duplication found was removed. Next, the titles and abstracts of the remaining articles were screened by ANS and NAA, and any articles that did not match the inclusion criteria were excluded. Lastly, the full text of the remaining articles were obtained and thoroughly checked to exclude any articles that did not meet the inclusion criteria (Figure 1).

The selection of articles to be included in the review had to be agreed by the two reviewers before proceeding to the data extraction phase. Disagreements were resolved by a third person (INM).

### 2.3. Data Extraction

Data extraction was performed by AnS and MFMM in a standardized manner with the use of a data collection form. Extracted data included (1) authors and year, (2) type of study, (3) objective, (4) molecular parameters measured, and (5) findings. Extracted data were tabulated to facilitate comparative analysis.

### 2.4. Quality Assessment

The quality of the included studies was evaluated by NAAN and NAA using SYRCLE’s risk of the bias tool for animal studies [15] and the Newcastle–Ottawa Scale (NOS) for human study [16] to ensure reliability. Any study rated as low quality was excluded, or their limitations were explicitly noted in this review.

### 2.5. Data Synthesis

Data synthesis was performed using a narrative synthesis approach. The key information extracted from the included studies is presented in a summary table. It summarized important data in a clear and organized manner to allow for a quick comparison and assessment of the characteristics, methods, and findings of the studies reviewed.

### 2.6. Handling Missing Data

For studies with missing or incomplete data, efforts were made to reach out to the corresponding authors for clarification. If no responses were received, the study was either excluded or analyzed using only the available data.

### 2.7. Ethical Considerations

Although this systematic review does not include direct human or animal experimentation, ethical research guidelines were followed. All studies considered were peer-reviewed publications that obtained ethical approval for their respective experiments and clinical trials.

## 3. Results

As shown in Figure 1, the initial search identified 513 potentially relevant articles. After screening the titles and abstracts, 463 articles were excluded, leaving 50 for full-text review. Of these, 22 studies were excluded, with 18 not assessing molecular parameters and 4 not related to osteoporosis, resulting in 28 articles included in the review (17–52).

Among these, there were 14 in vitro studies, 8 mixed in vitro and in vivo studies, and 6 in vivo studies. The primary cell lines used in in vitro studies were the pre-osteoblastic cell line (MC3T3-E1) and the bone stromal cell model (BSCM), while the main animal models were rats and mice. Additionally, only one cross-sectional study was conducted in postmenopausal women.

The results were categorized into three key areas of resveratrol’s effects: its impact on osteoblasts, osteoclasts, and osteoporosis models. This classification aids in a clearer understanding of the molecular mechanisms involved. The essential data from the included studies are summarized in a table (Table 2, Table 3 and Table 4).

### 3.1. Effects of Resveratrol on Osteoblast (Table 2)

Resveratrol, a polyphenolic compound, has garnered attention for its potential therapeutic effects on osteoblast activity and bone health, particularly in the context of osteoporosis. Numerous studies illustrate its multifaceted role in promoting osteoblast differentiation, survival, and proliferation, with significant implications for bone regeneration and the management of osteoporosis.

Resveratrol has been shown to augment osteogenic differentiation through the SIRT1/PI3K/AKT signaling pathway. Han et al. [17] demonstrated that resveratrol increased osteogenic markers such as Runx2, osterix (OSX), and osteocalcin (OCN) in bone marrow mesenchymal stem cells (BMSCs), with a dose-dependent effect on calcium deposition and alkaline phosphatase (ALP) activity. Similarly, Kuroyanagi et al. [18] reported that resveratrol suppressed M-CSF synthesis and osteoprotegerin (OPG) expression via PI3K/Akt signaling, further emphasizing the involvement of SIRT1 in osteoblast activation. This molecular pathway underlines the significance of resveratrol in promoting bone health.

In high-glucose environments, which typically impair osteoblast function, resveratrol has been found to restore osteoblast proliferation and differentiation. Hwang et al. [19] demonstrated that resveratrol mitigates the negative impact of hyperglycemia on osteoblasts by increasing ALP activity and the expression of key differentiation genes, including Collagen Type I, Transforming Growth Factor Beta 1 (TGF-β1), and OCN. These findings suggest that resveratrol can counteract osteoblast dysfunction associated with metabolic disorders, such as diabetes.

Another critical mechanism through which resveratrol promotes bone health is autophagy. Cai et al. [20] showed that resveratrol enhances autophagy-mediated proliferation and differentiation in MC3T3-E1 cells, with a concentration of 10 µmol/L producing the most significant effects on ALP activity and mineralization. The inhibition of autophagy reduced the benefits of resveratrol, further confirming that its effects are autophagy dependent. Wang et al. [21] also highlighted that resveratrol regulates autophagy pathways in postmenopausal osteoporosis rats, supporting osteoblastic differentiation and reversing estrogen deficiency-induced suppression of autophagy. These studies highlight the importance of autophagy in resveratrol’s ability to promote bone formation.

Oxidative stress is another key factor in osteoblast dysfunction, particularly in conditions like diabetes and aging. Xuan et al. [22] demonstrated that resveratrol activates nuclear factor erythroid 2-related factor 2 (NRF2) via the AKT/GSK3β/FYN axis to mitigate oxidative stress in osteoblasts under high-glucose conditions. This activation of NRF2 promotes osteoblast survival and differentiation, providing a protective mechanism against diabetic osteoporosis. Resveratrol’s antioxidant properties contribute significantly to its role in maintaining bone health under oxidative stress.

Resveratrol also modulates apoptosis and inflammatory cytokines, which are crucial for maintaining osteoblast function. Wang et al. [23] reported that resveratrol reverses osteogenic decline in BMSCs under inflammatory conditions by normalizing Hippo/YAP signaling. Additionally, He et al. [24] identified the TNF and forkhead box O (FoxO) signaling pathways as critical modulators of apoptosis and inflammation, which are regulated by resveratrol, enhancing osteoblast function.

The preservation of the stemness of mesenchymal stem cells (MSCs) is another important aspect of resveratrol’s role in bone health. Yahya et al. [25] found that resveratrol regulates Sex-Determining Region Y (SRY)-Box Transcription Factor 2 (Sox2) expression via SIRT1, maintaining MSC proliferation and self-renewal, which is crucial for sustained osteoblast generation. In the study by Hioki et al. [26], resveratrol activated SIRT1 to modulate excessive osteoblast migration induced by Insulin-Like Growth Factor 1, preventing uncontrolled bone remodeling. Similarly, Constanze et al. [27] reported that resveratrol activated SIRT1 and Runx2 to reverse TNF-β-induced inhibition of MSC osteogenic differentiation. This preservation of stem cell function suggests resveratrol’s broader therapeutic potential.

Resveratrol also influences microRNA regulation, which plays a significant role in osteoblast differentiation. Song et al. [28] showed that resveratrol modulates the MicroRNA-193a/Sirtuin 7 (miR-193a/SIRT7) axis, enhancing osteogenesis by suppressing miR-193a, which enables SIRT7 upregulation and the activation of NF-κB signaling. Zou et al. [29] found that resveratrol inhibits miR-320c, stabilizing the expression of Runx2 and promoting osteogenesis. These findings emphasize resveratrol’s ability to leverage microRNA pathways to regulate osteoblast function.

Furthermore, resveratrol protects osteoblasts from external stressors, such as dexamethasone-induced apoptosis and oxidative damage. Wang et al. [21] demonstrated that resveratrol activates the AMP-Activated Protein Kinase pathway to preserve mitochondrial integrity and enhance osteogenesis-related gene expression. Jiang et al. [30] further corroborated these findings, showing that resveratrol alleviates oxidative stress via the SIRT1/FoxO1 pathway, promoting osteoblast proliferation and differentiation.

Molecular pathways targeted by resveratrol further highlight its impact on osteoblast activity. Mei et al. [31] found that resveratrol modulates Extracellular Signal-Regulated Kinases 1 and 2 (ERK1/2) and c-Jun N-terminal Kinase (JNK) signaling to alleviate cadmium-induced inhibition of osteogenic differentiation. Yu et al. [32] identified the Mouse Double Minute 2/ Tumor Protein p53 (MDM2/p53) pathway as another mechanism through which resveratrol reverses p53-mediated inhibition of osteogenesis. These studies underscore resveratrol’s potential in targeting multiple molecular pathways to enhance osteoblast activity.

The estrogenic properties of resveratrol, possibly due to its chemical structure that is closely identical to that of diethylstilbesterol, a synthetic estrogen [33], may also contribute to its bone anabolic effects. Shah et al. [34] reported that resveratrol selectively activates the estrogen receptor-alpha (ERα) gene in osteoblasts, mimicking estrogen’s protective effects against postmenopausal bone loss. Liu et al. [35] further demonstrated that resveratrol suppresses Secreted Frizzled-Related Protein 1 (SFRP1), enhancing BMSC differentiation into osteoblasts and mitigating bone loss in ovariectomized models. These phytoestrogenic effects further support the therapeutic potential of resveratrol in osteoporosis management.

The multifaceted mechanisms of resveratrol including its regulation of osteoblast differentiation, autophagy, antioxidant properties, and modulation of signaling pathways, highlight its promising therapeutic potential for osteoporosis (Figure 2). However, further research, particularly clinical trials, is needed to fully understand its efficacy and therapeutic applications.

### 3.2. Effects of Resveratrol on Osteoclast (Table 3)

Multiple studies illustrate resveratrol’s significant inhibitory effects on osteoclast differentiation and activity through various molecular mechanisms. Lee et al. [36] demonstrated that oxyresveratrol, a derivative of resveratrol, inhibits osteoclastogenesis by suppressing the mitogen-activated protein kinase (MAPK) pathway, which includes p38, JNK, and ERK signaling. The study confirmed reduced expression of key osteoclast differentiation markers such as Nuclear Factor of Activated T Cells, Cytoplasmic 1 (NFATc1) and cathepsin K, alongside increased bone density in ovariectomized rats. Similarly, Poudel et al. [37] found that resveratrol prevents doxorubicin-induced osteoclast differentiation by enhancing antioxidant pathways, such as upregulating Superoxide Dismutase 1 (Sod1) and NRF2 expression while reducing the osteoclast markers RANK and Tartrate-Resistant Acid Phosphatase (TRAP). These findings suggest that resveratrol’s ability to inhibit osteoclast activity extends across various pathological conditions, highlighting its potential as a therapeutic agent for bone diseases such as osteoporosis.

Resveratrol modulates inflammation-driven osteoclast differentiation by targeting key signaling pathways. Xue et al. [38] reported that resveratrol enhances the expression of miR-181a-5p, which downregulates TRAF6 and TAK1 activity, crucial mediators of the NF-κB pathway. This inhibition results in reduced production of pro-inflammatory cytokines, such as TNF-α and IL-1β, and diminished osteoclast bone resorption activity. Additionally, Zhang et al. [39] observed that resveratrol suppresses the NADPH oxidase 4 (Nox4)/NF-κB pathway through the upregulation of miR-92b-3p, contributing to decreased osteoclast activity in estrogen deficiency-induced osteoporosis. These findings collectively emphasize the role of resveratrol in modulating inflammatory and oxidative stress pathways to protect against excessive bone resorption.

Yan et al. [40] highlighted that resveratrol alleviates hypoxia-induced osteoporosis by suppressing the ROS/HIF-1α signaling pathway, which promotes osteoclastogenesis under hypoxic conditions. Furthermore, Wei et al. [41] showed that resveratrol restores osteocyte autophagy and reduces oxidative stress-induced apoptosis in ovariectomized rats via the AMPK/JNK1 pathway. These studies underscore the compound’s dual functionality as an antioxidant and anti-inflammatory agent, effectively preserving bone integrity under stress-induced conditions (Figure 3 and Figure 4).

### 3.3. Role of Resveratrol in Osteoporosis Models and Pathways (Table 4)

Resveratrol has gained considerable attention for its therapeutic potential in osteoporosis, largely due to its multifaceted action on bone health through antioxidant properties, sirtuin modulation, and the regulation of bone marrow-derived mesenchymal stem cells (BMSCs). Several studies have revealed that resveratrol can alleviate bone damage by improving osteogenesis, reducing oxidative stress, and protecting against cellular apoptosis, thus supporting its role in bone health, especially under conditions such as menopause, aging, and metabolic disorders.

One of the major ways resveratrol exerts its effects on bone health is by restoring osteogenic function and alleviating oxidative stress-induced damage in bone cells. In research on acrylamide-induced skeletal toxicity, resveratrol demonstrated its antioxidant capacity by mitigating damage caused by oxidative stress [42]. This antioxidative mechanism is particularly relevant in conditions characterized by excessive oxidative damage, which is a key factor in bone degeneration. In addition, resveratrol has been shown to promote osteogenic differentiation in BMSCs derived from individuals with type 2 diabetes. By upregulating SIRT1 expression, resveratrol reduced apoptosis and stimulated bone formation, suggesting its ability to reverse cellular damage and enhance bone cell function in pathologies linked to oxidative stress and impaired osteogenesis [43]. These findings underline resveratrol’s ability to improve bone health by targeting oxidative stress pathways and promoting bone regeneration.

Further studies have explored resveratrol’s impact on bone mass, especially in animal models of osteoporosis and menopause. In ovariectomized rats, resveratrol increased bone mass and mineral density by enhancing the SIRT1/β-catenin pathway, which plays a critical role in bone formation. This result is particularly relevant as it parallels human studies in which higher serum levels of SIRT1 correlate with better bone health, particularly in postmenopausal women [44]. The activation of SIRT1, a sirtuin protein involved in regulating cellular metabolism and stress resistance, appears to be a crucial mechanism by which resveratrol mitigates bone loss, particularly in conditions of hormonal imbalance, such as during menopause. Resveratrol’s ability to modulate the SIRT1/β-catenin axis further emphasizes its potential as a therapeutic agent in combating osteoporosis and maintaining bone integrity during periods of hormonal decline. Similarly, resveratrol counteracted indoxyl sulfate-induced inhibition of osteoblastogenesis by activating the SIRT1 pathway [45]. In a study by Shi et al. [46], resveratrol ameliorated bone loss in ovariectomized mice by reducing SFRP1 levels, thereby activating Wnt/β-catenin signaling and promoting osteogenesis.

Moreover, resveratrol shows promise in addressing bone marrow adiposity (BMAT), a condition associated with osteoporotic fragility and aging. Research has demonstrated that resveratrol inhibits adipogenic differentiation while promoting osteoblastic differentiation in human BMSCs, particularly from aged individuals. This dual effect reduces BMAT accumulation, a key contributor to bone loss, and restores bone homeostasis [47]. In addition, resveratrol’s influence on senescence-associated pathways, such as the FoxO1/SIRT1 axis, has been shown to regulate the differentiation of skeletal stem cells into osteoblasts, further supporting its role in mitigating aging-related bone loss [48]. The ability of resveratrol to modulate BMAT and target senescence pathways underscores its potential in treating aging-related bone diseases and metabolic disorders, which are closely tied to reduced bone mass and strength.

Resveratrol also proves beneficial in conditions involving chronic energy deficiency and oxidative damage. In glucocorticoid-induced osteonecrosis, resveratrol activates SIRT3, another sirtuin protein, which helps reduce oxidative stress and protect bone cells from damage [49]. Additionally, resveratrol suppresses the ROS/HIF signaling pathway in high-altitude-induced osteoporosis, indicating its broad applicability in various forms of bone degeneration associated with oxidative stress and energy imbalances [40]. These findings highlight resveratrol’s ability to intervene in diverse pathophysiological conditions that contribute to bone loss, further reinforcing its therapeutic potential in the management of osteoporosis and related bone diseases.

Overall, resveratrol’s effects on osteoblasts and osteoclasts are largely consistent across different cell lines and animal models, demonstrating its efficacy in promoting bone formation and inhibiting bone resorption (Figure 5). The reproducibility of its effects in various pathological conditions, such as oxidative stress, metabolic disorders, and estrogen deficiency, further reinforces its therapeutic potential.

**Table 2 ijms-26-02893-t002:** Effects of resveratrol on osteoblasts.

Author and Year	Types of Study	Objective	Molecular Parameters	Findings
Kuroyanagi et al., 2023 [18]	In vitro (MC3T3-E1 cells)	To investigate resveratrol’s effects on M-CSF synthesis.	M-CSF, OPG, Akt, PI3-kinase, and SIRT1	Resveratrol reduces bFGF-induced M-CSF synthesis and OPG mRNA expression through the PI3-kinase/Akt pathway and SIRT1 activation in osteoblasts.
Han et al., 2024 [17]	In vivo (mice) and in vitro (BMSCs)	To study resveratrol’s effect on osteogenic differentiation of BMSCs.	Runx2, OSX, OCN, calcium deposition, ALP activity, SIRT1, and PI3K/AKT	Resveratrol enhances osteogenic differentiation in osteoporosis mice through the SIRT1/PI3K/AKT pathway, improving various bone parameters in vivo and in vitro.
Hwang et al., 2024 [19]	In vitro (MC3T3-E1 and hPDLF cells)	To explore resveratrol’s effect on bone formation under high-glucose conditions.	ALP activity, Col 1, TGF-β1, ALP, OCN, and RUNX2	Resveratrol facilitates osteoblast function under high-glucose conditions, showing stage-specific effects on osteogenic differentiation in MC3T3-E1 cells and hPDLF cells.
Cai et al., 2023 [20]	In vitro (MC3T3-E1 cells)	To investigate resveratrol’s effects on proliferation and differentiation in MC3T3-E1 cells.	Runx2, OCN, and autophagy markers (LC3II/LC3I, and p62)	Resveratrol promotes osteogenic differentiation in MC3T3-E1 cells by activating autophagy.
He et al., 2024 [24]	In vitro (MC3T3-E1 cells)	To explore the mechanism by which resveratrol promotes the proliferation and differentiation of MC3T3-E1 cells.	Runx2, OPG, TNF, IL6, CASP3, and apoptosis-related proteins	Resveratrol promotes osteogenic differentiation by inhibiting apoptosis and regulating apoptosis-related proteins such as TNF and IL6, with potential implications for osteoporosis treatment.
Wang et al., 2022 [23]	In vitro (BMSCs)	To examine the effect of resveratrol on osteogenic differentiation in inflammatory conditions.	Inflammatory cytokines (IL-6, Mmp-9, and Il-1β), Hippo/YAP signaling, and YAP expression	Resveratrol reverses osteogenic decline by modulating inflammatory cytokines and restoring Hippo/YAP signaling in BMSCs.
Yahya et al., 2022 [25]	In vitro (Human BM-MSCs)	To investigate resveratrol’s effect on maintaining the stemness of human BM-MSCs.	SIRT1, Sox2, and stemness markers (CD73, CD90, and CD105)	Resveratrol maintains the stemness of human iliac BM-MSCs by activating SIRT1 and regulating Sox2 expression, promoting cell proliferation and reducing apoptosis and senescence.
Xuan et al., 2022 [22]	In vitro (Osteoblasts)	To explore resveratrol’s effect on osteoblast dysfunction under high glucose.	NRF2, AKT, GSK3β, FYN, and oxidative stress markers	Resveratrol alleviates osteoblast dysfunction under high glucose by activating NRF2 through the AKT/GSK3β/FYN axis, suppressing oxidative stress.
Liu et al., 2021 [35]	In vitro (MC3T3-E1 cells)	To explore resveratrol’s effect on osteoblast proliferation via GATA-1 and autophagy.	GATA-1, AMPKα, and autophagy markers	Resveratrol promotes osteoblast proliferation by activating GATA-1 and autophagy, with AMPKα acting as an upstream regulator.
Hioki et al., 2020 [26]	In vitro (MC3T3-E1 osteoblast-like cells)	To investigate the effects of resveratrol on IGF-I-induced osteoblast migration.	p44/p42 MAPK, Akt, SIRT1, IGF-I, and osteoblast migration	Resveratrol suppresses IGF-I-induced osteoblast migration via SIRT1 activation and the attenuation of the p44/p42 MAPK pathway.
Constanze et al., 2020 [27]	In vitro (MSC cultures)	To investigate the effect of TNF-β on MSC osteogenic differentiation and its reversal by resveratrol.	NF-κB, Sirt1, Runx2, β1-integrin, and osteogenic differentiation markers	Resveratrol reverses TNF-β-induced suppression of MSC osteogenesis by activating SIRT1 and Runx2, involving NF-κB modulation.

**Table 3 ijms-26-02893-t003:** Effects of resveratrol on osteoclast.

Authors and Year	Types of Study	Objective	Molecular Parameters	Findings
Lee et al., 2024 [36]	In vivo (OVX rat model) and in vitro (RAW 264.7 cell line)	To investigate the effect of oxyresveratrol on osteoclast differentiation and bone density in osteoporosis.	TRAP, MAPK (p38, JNK, and ERK), NFATc1, Cathepsin K, Bone mineral density (BMD), DPD, and TRAP activity	Oxyresveratrol inhibits osteoclast differentiation and MAPK phosphorylation and increases bone density in ovariectomized rats, suggesting a potential therapeutic use for osteoporosis.
Wei et al., 2023 [41]	In vivo (ovariectomized rat) and in vitro (MLO-Y4 osteocyte cells)	To investigate the protective effects of resveratrol on osteocytes against oxidative stress and apoptosis.	AMPK, JNK1, Beclin-1/Bcl-2 complex, autophagy markers, and osteocyte apoptosis markers	Resveratrol activates autophagy, reduces apoptosis, and protects osteocytes through the AMPK/JNK1-mediated pathway in ovariectomized rats and MLO-Y4 cells.
Poudel et al., 2022 [37]	In vitro (RAW 264.7 cells) and in vivo (zebrafish model)	To study resveratrol’s effect on doxorubicin-induced osteoclast differentiation and osteotoxicity.	Oc-stamp, RANK, TRAP, Ctsk, NRF2, SOD1, MitoTEMPO, and locomotion parameters	Resveratrol reverses DOX-induced osteoclast differentiation, reduces oxidative stress, and improves locomotion in zebrafish, suggesting protective effects against chemotherapy-induced bone loss.
Xue et al., 2023 [38]	In vitro (RAW 264.7 cell line)	To evaluate the effect of resveratrol on LPS-induced osteoclast differentiation through miR-181a-5p.	miR-181a-5p, TRAF6, P-IκB-α, NF-κB65, CTSK, MMP-9, and TRAP	Resveratrol inhibits osteoclast differentiation through miR-181a-5p-mediated suppression of the TRAF6/TAK1 pathway in LPS-stimulated RAW 264.7 cells.
Yan et al., 2022 [40]	In vivo (Wistar rat model) and in vitro (BMSCs and RAW264.7 cells)	To investigate resveratrol’s effect on high-altitude hypoxia-induced osteoporosis.	ROS, HIF-1α, RUNX2, ALP, OCN, CTX-I, TRAP, BMD, and BMSC proliferation and differentiation	Resveratrol mitigates osteoporosis caused by high-altitude hypoxia by suppressing the ROS/HIF-1α signaling pathway to inhibit osteoclastogenesis.
Zhang et al., 2020 [39]	In vivo (OVX rat model) and in vitro (bone tissue analysis)	To investigate resveratrol’s effect on estrogen deficiency-induced osteoporosis via NADPH oxidase and the NF-κB pathway.	miR-92b-3p, Nox4, NF-κBp65, BMP2, Smad7, RUNX2, Cathepsin K, BMD, and TRAP	Resveratrol alleviates estrogen deficiency-induced osteoporosis by enhancing miR-92b-3p expression, inhibiting Nox4/NF-κB signaling, and decreasing osteoclast activity.

**Table 4 ijms-26-02893-t004:** Role of resveratrol in osteoporosis models and pathways.

Author and Year	Types of Study	Objective	Molecular Parameters	Findings
Zhang H. et al., 2024 [42]	In vivo (mice)	To investigate the skeletal toxicity of acrylamide (AA) and the protective effect of resveratrol (RVT).	RT-qPCR and histopathological analysis	AA induced osteogenesis and cartilage damage. RVT restored bone function and reduced apoptosis and oxidative stress in osteogenesis/cartilage.
Deng et al., 2023 [43]	In vitro (BMSC from T2DM patients)	To examine SIRT1 expression on osteogenic differentiation of BMSCs in type 2 diabetes mellitus (T2DM).	SIRT1 expression and BMSC osteogenic differentiation markers	Resveratrol increased SIRT1 expression, reduced BMSC apoptosis, and promoted osteogenic differentiation in T2DM patients.
Wang et al., 2023 [44]	Animal (ovariectomised rats) and cross-sectional study (postmenopausal women)	To examine the effects of resveratrol on bone mass in ovariectomized rats and the association of SIRT1 rs7896005 SNP with bone mass in women.	SIRT1, β-catenin, OPG, RANKL, and bone mineral density (BMD)	Resveratrol increased SIRT1, β-catenin, and BMD. Women with the A allele of SIRT1 rs7896005 had lower bone mass.
Chen et al., 2021 [49]	Animal study (rats)	To investigate the role of SIRT3 in glucocorticoid-induced osteonecrosis of the femoral head (GIONFH) and resveratrol’s effect.	SIRT3, oxidative stress markers, and the AMPK/PGC-1α/SIRT3 pathway	SIRT3 expression was reduced in GIONFH rats. Resveratrol activated SIRT3 and reduced oxidative stress, promoting osteogenic differentiation in BMSCs.
Yan et al., 2022 [40]	Animal and in vitro study (rats and BMSCs)	To investigate the effect of resveratrol on high-altitude hypoxia-induced osteoporosis.	ROS, HIF-1α, RUNX2, ALP, OCN, CTX-I, and TRAP	Resveratrol prevented high-altitude hypoxia-induced osteoporosis by promoting osteoblastogenesis and inhibiting osteoclastogenesis via ROS/HIF-1α suppression.
Ameen et al., 2020 [48]	Animal study (rats)	To investigate resveratrol’s therapeutic effect on aging-dependent male osteoporosis.	FoxO1, SIRT1, RANKL, OPG, and bone-specific markers	Resveratrol restored age-related osteoporotic changes by modulating inflammation, oxidative stress, and the gene expression of FoxO1 and SIRT1.
Ali et al., 2020 [47]	In vitro study (human BMSCs)	To examine resveratrol’s effect on adipocyte differentiation and cellular senescence in bone marrow stromal stem cells (BMSCs).	Adipocyte and osteoblast differentiation markers, ROS, SIRT1, FAK, and AKT	Resveratrol inhibited adipocyte differentiation and senescence, enhancing osteoblast differentiation in human BMSCs.
Louvet et al., 2020 [50]	Animal study (mice)	To investigate the role of SIRT1 in bone marrow adiposity (BMA) and its relationship with osteogenesis and adipogenesis.	SIRT1, Runx2, FoxO1, and acetylation levels	SIRT1 activation by resveratrol reduced BMA and promoted osteogenesis in an anorexia nervosa model by modulating the acetylation levels of Runx2 and FoxO1.
Shi et al., 2022 [46]	In vivo (OVX mice model)	To investigate the effect of resveratrol on osteoporosis in OVX mice through SFRP1-mediated osteogenesis.	SFRP1, ALP, osteogenesis-associated genes, and bone mineral density (BMD)	Resveratrol ameliorates bone loss in OVX mice by upregulating osteogenesis-associated genes and reducing SFRP1 levels, potentially useful for treating postmenopausal osteoporosis.
Liu et al., 2020 [45]	In vivo (nephrectomy mouse model) and in vitro (MC3T3-E1 cells)	To study the protective role of resveratrol against indoxyl sulfate (IS)-induced inhibition of osteoblastogenesis.	Runx2, ERK, p38 MAPK, AhR, SIRT1, and osteoblast differentiation markers	Resveratrol counteracts IS-induced inhibition of osteoblastogenesis through the AhR/MAPK/SIRT1 signaling pathway in vitro and in vivo.

## 4. Discussion

Resveratrol, a polyphenolic compound, has gained significant attention for its potential therapeutic role in the management of osteoporosis, particularly through its molecular actions on various cellular and metabolic pathways. Recent studies have highlighted its efficacy in protecting against bone loss induced by different pathophysiological conditions, including aging, high-altitude hypoxia, and glucocorticoid use. One common mechanism is the modulation of oxidative stress, a critical factor in bone homeostasis. Research has shown that resveratrol exerts antioxidant effects by reducing reactive oxygen species (ROS) and activating sirtuin pathways, which play a key role in osteogenesis [40,42,48].

In models of osteoporosis induced by glucocorticoids and high-altitude hypoxia, resveratrol has demonstrated the ability to ameliorate bone loss and enhance bone mineral density (BMD). Specifically, resveratrol inhibits osteoclastogenesis whilst promoting osteoblastogenesis through pathways like ROS/HIF-1α signaling and the AMPK/PGC-1α/SIRT3 axis [40,50]. In hypoxic conditions, resveratrol has been shown to reverse the reduction in BMD by modulating key osteogenic markers such as alkaline phosphatase (ALP), osteocalcin (OCN), and runt-related transcription factor 2 (RUNX2) [40]. These findings suggest that resveratrol’s therapeutic potential may extend to various forms of osteoporosis, including those linked to environmental stressors like high-altitude hypoxia.

The role of sirtuins, particularly SIRT1, in the osteogenic differentiation of mesenchymal stem cells (MSCs) is also a focal point of resveratrol research since the polyphenol is a known SIRT1 activator. In patients with type 2 diabetes mellitus, resveratrol enhanced the expression of SIRT1, thereby improving the osteogenic potential of bone marrow MSCs [43]. Similarly, in male osteoporosis models, resveratrol’s effects on the FoxO1/SIRT1/RANKL/OPG signaling pathway have been linked to its capacity to prevent the progression of bone loss through anti-inflammatory and antioxidant mechanisms [48]. Resveratrol also activates the focal adhesion kinase (FAK) and Akt pathways to inhibit adipogenic differentiation and promote osteoblastic differentiation in human BMSCs [47]. This modulation of cellular differentiation is crucial in preventing the accumulation of bone marrow adipose tissue (BMAT), which is often associated with osteoporosis and reduced bone strength.

Furthermore, resveratrol’s influence on gene expression and cellular activities is reflected by in its ability to modulate the acetylation of transcription factors such as Runx2 and FoxO1, which are critical for osteoblast differentiation [50]. This action emphasizes the potential of resveratrol as a therapeutic agent for osteoporosis, particularly in conditions where sirtuin activity is impaired, such as chronic energy deficiency or aging. Additionally, the ability of resveratrol to regulate the expression of OPG and receptor activator of NF-κB ligand (RANKL) provides further insights into its protective effects on bone metabolism [43,44].

Resveratrol’s multifaceted mechanism of action, involving oxidative stress regulation, sirtuin activation, and the modulation of key signaling pathways, positions it as a promising candidate for the treatment of osteoporosis. Its potential benefits across different forms of osteoporosis, regardless of whether it was induced by aging, hypoxia, or glucocorticoids, made it an attractive option for future clinical applications.

Resveratrol has also drawn attention to its potential in protecting against bone loss by inhibiting osteoclastogenesis and reducing bone resorption. Osteoclasts are the primary cells responsible for bone resorption, and their overactivation is a major factor in the development of osteoporosis. Recent studies have demonstrated that resveratrol exerts inhibitory effects on osteoclast differentiation and activity through various molecular mechanisms [38,51], positioning it as a promising therapeutic agent for bone diseases.

Osteoclast differentiation is regulated by several signaling pathways, including the mitogen-activated protein kinase (MAPK) pathway and the NF-κB pathway. Lee et al. [36] showed that oxyresveratrol, a resveratrol derivative, inhibits osteoclastogenesis by suppressing MAPK signaling, particularly the p38, JNK, and ERK pathways. This suppression reduces the expression of key osteoclast markers like NFATc1 and cathepsin K, leading to increased bone density in ovariectomized rats, a common model of postmenopausal osteoporosis. Similarly, Poudel et al. [37] found that resveratrol inhibits osteoclast differentiation in doxorubicin-induced osteoporosis by enhancing antioxidant pathways, such as upregulating Sod1 and NRF2, and downregulating osteoclast markers like RANK and TRAP. These findings indicate that resveratrol’s effect on osteoclastogenesis is applicable across various pathological conditions, making it a promising candidate for osteoporosis therapy.

In addition to its effects on MAPK signaling, resveratrol also modulates inflammation-driven osteoclast differentiation. Xue et al. [38] reported that resveratrol increases the expression of miR-181a-5p, which targets TRAF6 and TAK1, key components of the NF-κB signaling pathway. By inhibiting NF-κB signaling, resveratrol reduces the production of pro-inflammatory cytokines, such as TNF-α and IL-1β, which are known to stimulate osteoclast differentiation. Zhang et al. [39] further observed that resveratrol suppresses the Nox4/NF-κB pathway through upregulation of miR-92b-3p, resulting in decreased osteoclast activity in estrogen-deficiency-induced osteoporosis. These studies highlight the role of resveratrol in modulating inflammatory pathways, which are critical in the pathogenesis of osteoporosis.

Resveratrol also exerts protective effects in conditions involving oxidative stress and hypoxia, both of which are associated with increased osteoclastogenesis. Yan et al. [40] demonstrated that resveratrol mitigates hypoxia-induced osteoporosis by suppressing the ROS/HIF-1α signaling pathway. Hypoxia and ROS promote osteoclast differentiation, and resveratrol’s ability to inhibit this pathway protects against bone loss under hypoxic conditions. Wei et al. [41] showed that resveratrol restores osteocyte autophagy and reduces oxidative stress-induced apoptosis in ovariectomized rats by modulating the AMPK/JNK1 pathway. These findings underscore resveratrol’s dual function as an antioxidant and anti-inflammatory agent capable of protecting bone integrity under stress-induced conditions.

The findings from these studies suggest that resveratrol could be a valuable therapeutic agent for osteoporosis, as it inhibits osteoclast differentiation, reduces bone resorption, and counteracts oxidative stress and inflammation. However, its clinical application requires further research to establish optimal dosages and formulations, bioavailability, and long-term safety. As a natural product, resveratrol offers several advantages, including better compatibility with the human body, which leads to fewer side effects. Many natural products are metabolized efficiently, reducing toxicity risks [52]. Furthermore, they often exhibit multi-target activity, making them effective against complex diseases such as cancer, diabetes, and neurodegenerative disorders [53].

For future directions, we should focus on optimizing the bioavailability of resveratrol through nanotechnology-based delivery systems or structural modifications to enhance its effectiveness [54]. Additionally, combining resveratrol with existing pharmacological agents, such as bisphosphonates or selective estrogen receptor modulators (SERMs), could enhance treatment efficacy. Combination therapies may offer a more comprehensive approach to managing osteoporosis by synergizing the effects of resveratrol and conventional medications.

Recent advancements in resveratrol research focus on structural modifications and nanotechnology-based delivery systems to improve its chemical stability, solubility, bioavailability, and metabolic profile. Methoxylation enhances lipophilicity and metabolic stability, while hydroxylation increases water solubility and antioxidant activity. Phospholipid complexes improve solubility and inhibit glucuronidation, whereas liposomal co-delivery with polyphenols enhances stability and therapeutic potential. These modifications not only enhance resveratrol’s physicochemical properties but also influence its molecular signaling pathways, potentially increasing its efficacy in disease treatment.

All these modifications increase the bioavailability of resveratrol in the bone microenvironment, which can significantly influence its molecular mechanisms [55,56]. The impacts on molecular mechanisms include enhanced activation of SIRT1 and Wnt/β-Catenin signaling, as well as a stronger inhibition of RANKL-induced NF-κB, MAPK, and HIF-1α signaling. Therefore, an increase in resveratrol’s bioavailability in the bone microenvironment would intensify its beneficial molecular effects, leading to enhanced bone formation, reduced bone resorption, and improved bone quality [57].

Overall, these strategies address resveratrol’s limitations, making it more viable for clinical applications [11,58,59,60]. Since most studies on resveratrol’s bone-protective effects are preclinical, large-scale, long-term clinical trials are needed to confirm its efficacy and safety in osteoporosis patients [11].

## 5. Conclusions

In conclusion, resveratrol demonstrates significant promise as a natural therapeutic agent for osteoporosis due to its ability to regulate osteoblast, osteoclast, and osteocyte activities. Resveratrol promotes osteoblastogenesis by activating the PI3K/Akt, SIRT1, AMPK, and GATA-1 pathways, while its activation of the Hippo/YAP pathway reduces inflammation, and the NRF2 pathway mitigates oxidative stress. In osteoclasts, resveratrol inhibits osteoclastogenesis by suppressing the MAPK and TRAF6/TAK1 pathways while also reducing oxidative stress and inflammation through the suppression of the ROS/HIF-1α and Nox4/NF-κB pathways, respectively. Additionally, resveratrol enhances osteocyte function by promoting autophagy via AMPK/JNK1 activation, further contributing to bone homeostasis. Although further research, particularly clinical trials, is needed to substantiate its long-term efficacy and optimal dosage, the current evidence underscores resveratrol’s potential as an effective adjunct in osteoporosis management, particularly in conditions involving oxidative stress, hormonal changes, and metabolic dysfunction.

## Figures and Tables

**Figure 1 ijms-26-02893-f001:**
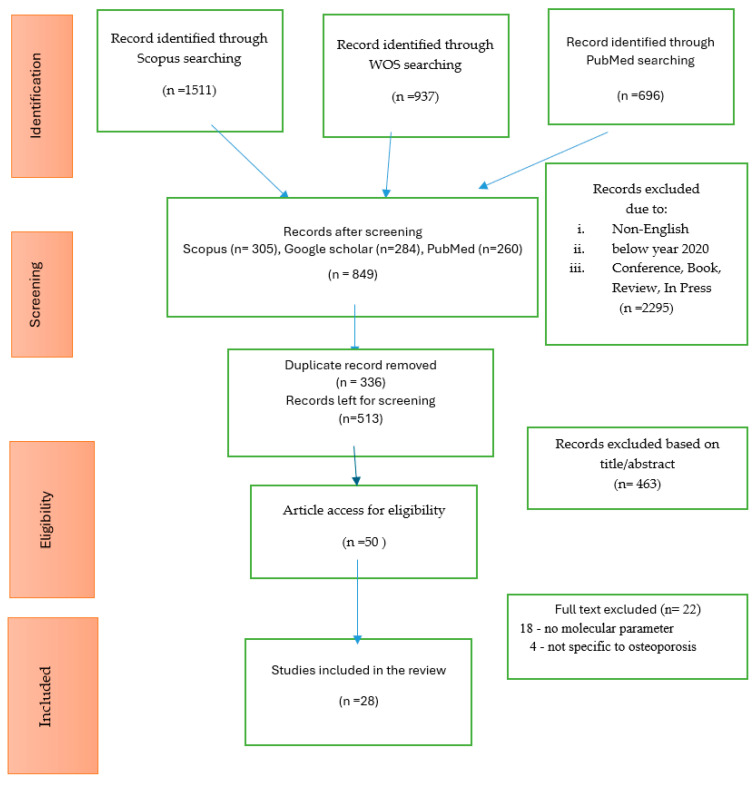
Flow diagram of the proposed searching study.

**Figure 2 ijms-26-02893-f002:**
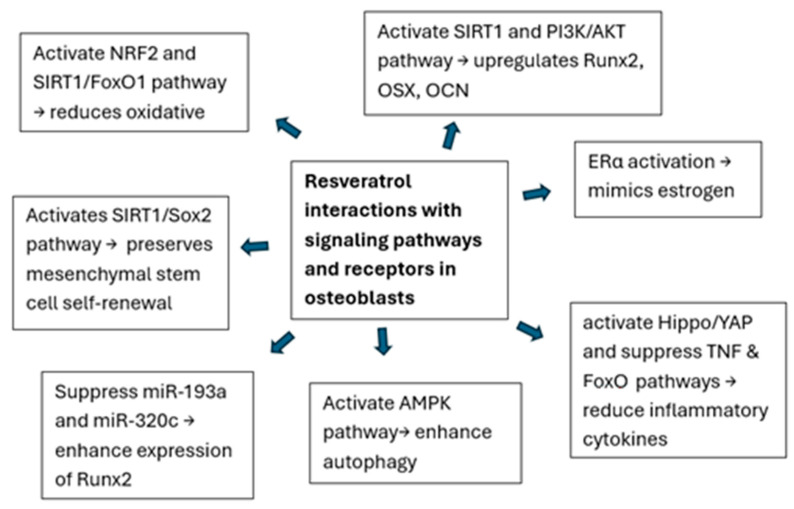
Illustration of the interactions of resveratrol with signaling pathways and receptors in osteoblasts.

**Figure 3 ijms-26-02893-f003:**
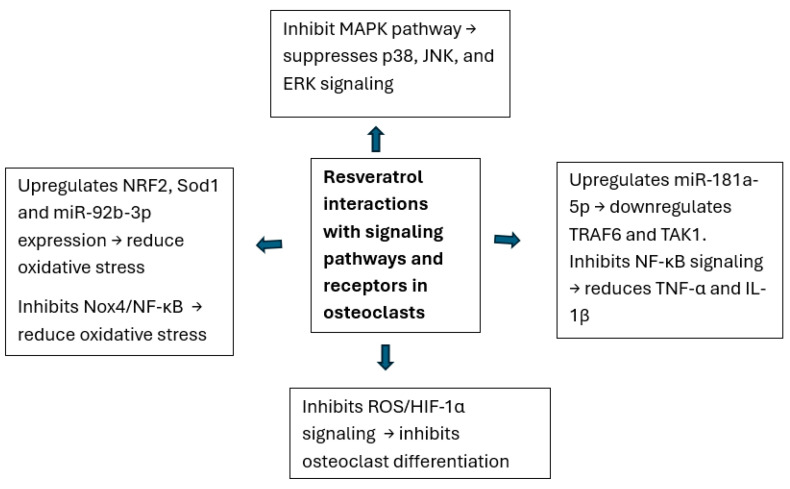
Illustration of the interactions of resveratrol with signaling pathways and receptors in osteoclasts.

**Figure 4 ijms-26-02893-f004:**
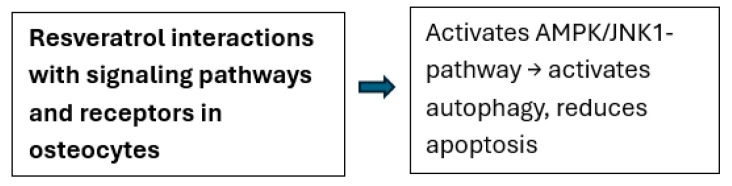
Illustration of the interactions of resveratrol with signaling pathways and receptors in osteocytes.

**Figure 5 ijms-26-02893-f005:**
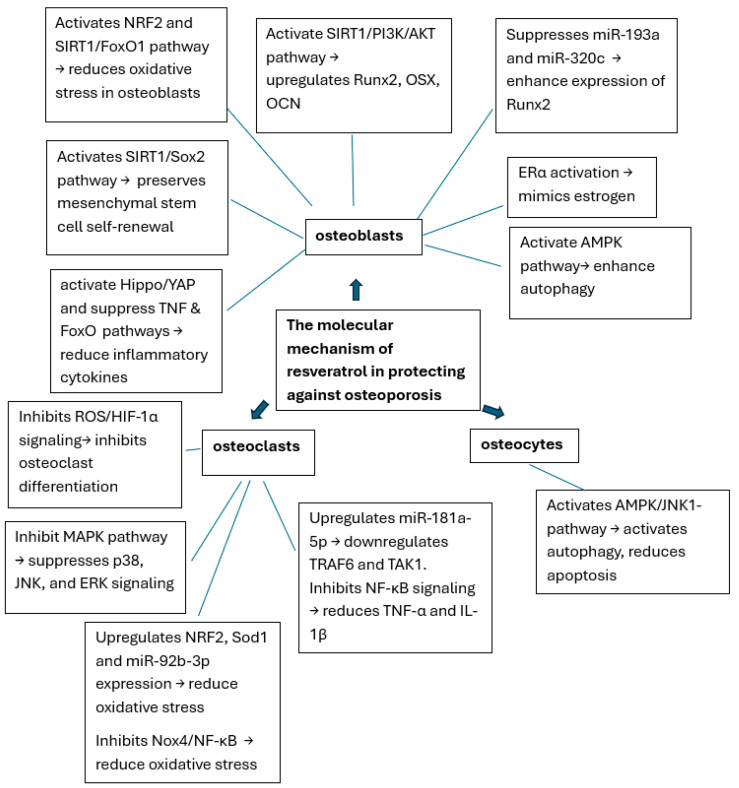
Graphical abstract of the molecular mechanisms of resveratrol in protecting against osteoporosis.

**Table 1 ijms-26-02893-t001:** PIPOC Framework.

	Inclusion	Exclusion
Population	Cell line Animal model Human	-
Intervention	Resveratrol	-
Comparison	Group not receiving resveratrol Positive control and negative control groups	-
Outcome	Molecular parameters related to mechanisms of protection against osteoporosis	Did not measure molecular parameters
Study Type	In vitro studies, animal studies, randomized controlled studies, case–control studies, cohort studies	case reports, editorials, communications, reviews, meta-analysis

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
