# Peer review of "A Systematic Review on the Molecular Mechanisms of Resveratrol in Protecting Against Osteoporosis"

_ijms, 2025, doi:10.3390/ijms26072893_

Round 1

Reviewer 1 Report

Comments and Suggestions for Authors

The review highlights specific molecular pathways through which resveratrol exerts its osteoprotective effects, such as enhancing osteoblast differentiation and inhibiting osteoclastogenesis. This level of detail adds valuable insights into the biological effects of resveratrol, which is crucial for understanding its potential as a treatment for osteoporosis. However, while it effectively explores molecular mechanisms, it does not fully address broader clinical aspects, such as optimal dosages, long-term effects, and potential interactions with medications or other osteoporosis treatments. While the paper provides a systematic overview of the mechanisms by which resveratrol may protect against osteoporosis, further research and careful interpretation of results are necessary before drawing definitive conclusions or making clinical recommendations.

Reviewer 2 Report

Comments and Suggestions for Authors

Comments:

Line 144: Its not clear why authors excluded such a large no of articles (513) and selected only 28 articles. 

152: its impact on osteoblasts, osteoclasts, and osteoporosis models: Should include impact on osteocytes as well.

Provide an individual illustration detailing the molecular interactions of resveratrol and cellular receptors in osteoblasts, osteoclast and osteocytes, respectively which is primarily important to understand the signaling concept. 

Provide any advancements in structural/chemical modification to improve chemical instability, poor water solubility, low bioavailability and rapid metabolism of resveratrol and how do they affect the molecular signaling mechanism. 

The paper lacks significant reference works. 

Also provide an individual illustration to describe the molecular signaling mechanism of Resveratrol in all three cells during osteoporosis condition.  

Comments on the Quality of English Language

Fine

Reviewer 3 Report

Comments and Suggestions for Authors
  1. There have been many review papers on the improvement of osteoarthritis by resveratrol (such as, The effects ofresveratrol in animal models of primary osteoporosis: a systematic review and meta-analysis, Microbiota and Resveratrol: How Are They Linked to Osteoporosis?, Protective effect and possible mechanisms of resveratrol in animal models of osteoporosis: A preclinical systematic review and meta-analysis etc). Please explain the difference between this paper and it. What are its features?
  2. Figure 1: The font size needs to be larger for clearer viewing.
  3. Line 181: “OC”or “OCN” Full name???
  4. MC3T3-E1??? BSCM???
  5. "Nrf2" needs to standardize case (NRF2).
  6. Some referencecitation formats are not uniform (e.g., parentheses are missing after the year). They need to be unified into the [author, year] format.
  7. Excluding studies on "combination with other drugs" may be unreasonable, as combination therapy may reveal the synergistic mechanism of resveratrol.
  8. Line 297:”......bone formation (Wang et al., 2023)”
  9. What are the advantages of resveratrol as a natural active ingredient in improving osteoporosis?What should we watch for in the future?
  10. How consistent is the effect of resveratrol across cell lines or animal models?
  11. The structural framework of the manuscript is too simple, and further refinement is recommended.
Comments on the Quality of English Language

could be improved

Round 2

Reviewer 2 Report

Comments and Suggestions for Authors

Authors did not address all my comments. 

C 3: Provide an individual illustration detailing the molecular interactions of resveratrol and cellular receptors in osteoblasts, osteoclast and osteocytes, respectively which is primarily important to understand the signaling concept.
Author Response: We have included illustrations of the molecular interactions in all three cell types as a single figure in the manuscript. Adding separate figures for each cell type would result in too many figures

New Comment: The said illustrations describe molecular signaling mechanism, but not receptor interaction. Provide illustrations for RECEPTOR Interaction of Res in all three cells, which is very important how they trigger and regulate biological cues. 

C 4. Provide any advancements in structural/chemical modification to improve chemical instability, poor water solubility, low bioavailability and rapid metabolism of resveratrol and how do they affect the molecular signaling mechanism.
Author response: The statement below on advancements in structural/chemical modification to improve chemical instability, poor water solubility, low bioavailability and rapid metabolism of resveratrol has been added to the discussion section.
“Recent advancements in resveratrol research focus on structural modifications and nanotechnology-based delivery systems to improve its chemical stability, solubility, bioavailability, and metabolic profile. Methoxylation enhances lipophilicity and metabolic stability, while hydroxylation increases water solubility and antioxidant activity. Phospholipid complexes improve solubility and inhibit glucuronidation,
whereas liposomal co-delivery with polyphenols enhances stability and therapeutic potential. These modifications not only enhance resveratrol’s physicochemical properties but also influence its molecular signaling pathways, potentially increasing its efficacy in disease treatment. Overall, these strategies address resveratrol’s limitations, making it more viable for clinical applications (Li 2021, Huang 2022, Chopra 2022, Tian et al 2020)”.

New comments: The authors did not well explain how do structural/chemical modification affect the molecular signaling mechanism, simply wrote a sentence like "These modifications not only enhance resveratrol’s physicochemical properties but also influence its molecular signaling pathways". The major focus of this review is "Molecular Mechanisms of Resveratrol", hence these details should be elaborated clearly in text as well as in Figures.

New comments: There are only two figures in this manuscript, and one of them are Flow diagram of the study. 

New comments: 5. Conclusions is poorly summarized, lacking to highlighted major cues in pathways. Write this section mainly focusing on the major aim of this paper. now the authors used general term "various molecular mechanisms"

New comments: various molecular mechanisms including "antioxidant effects"- Not correct, Antioxidant effects are not molecular mechanism.

New comments:  sirtuin activation, and modulation of inflammatory pathways: These are pathways, not molecular mechanism. Be specific in conclusion. 

C5.The paper lacks significant reference works.
Author response: We have added more recent and significant references. Among the references are:....

New Comments: The author did not add the references in the reference section as they said. There are only 52 reference as original submission. The authors are inattentive. 

Additional Comment:

In abstract, the author described general statement like "osteoblast differentiation through the upregulation of RUNX2, ALP, and osteocalcin...." and "by suppressing the RANKL signaling pathway....", which are biomarkers (Not exactly molecular mechanism) in bone homeostasis, however, the abstract of this review should focus on the Molecular Mechanisms of Resveratrol, which means how Res influences these biomarkers through various molecular signals in osteoporosis compared to the normal state, to highlights the specific role of Res's behaviour in diseased condition for therapeutic approach.  

Comments on the Quality of English Language

Need to check the Scientific grammar thoroughly. 

Author Response

Please see the attachment, Thank you

Reviewer 3 Report

Comments and Suggestions for Authors

There are no further comments

Author Response

Thank you 

Round 3

Reviewer 2 Report

Comments and Suggestions for Authors

No more comments